# Mapping Data Governance Requirements Between the European Union's AI Act and ISO/IEC 5259: A Semantic Analysis*

Kuruvilla George Aiyankovil[1,*,†], Julio Hernandez[2,†] and Dave Lewis[3,†]

[1]*Trinity College Dublin*
[2]*Adapt Center, Ireland*
[3]*Trinity College Dublin*

## Abstract

The rapidly evolving landscape of artificial intelligence (AI) has made regulatory frameworks essential to guide the development, deployment, and usage of AI technologies responsibly. Recently, the European Union (EU) has approved the AI Act to address these needs, laying out a set of requirements for AI systems. Similarly, the EU published a request for harmonized standards that would support implementation of the AI Act across a number of topics related to trustworthy AI and AI quality and management. One source for such European Harmonised Standards are International Standards, and ISO/IEC JTC1 SC42 has a number of standards published and in development that may be appropriate, but official analysis shows some gaps that require additional features for existing standards. It is not clear that near term modifications to existing standards will satisfy all the requirement of the AI Act given the complexity and lack of state of the art in many areas, especially in novel area such as protection of fundamental rights.We propose therefore the use of semantic web vocabularies to track the mappings of AI Act requirements that will enable the progressive tracking to third party guidelines, standards and specification. In particular, we demonstrate this approach by producing a requirement analysis of Article 10 of the AI Act on Data Governance and map it to the relevant provisions of the SC42 standards 5259 on Data Quality for Machine Learning. This study conducts a semantic analysis of the EU's AI Act and ISO/IEC 5259 requirements, utilizing the Simple Knowledge Organization System (SKOS) ontology to map concepts between these two frameworks. We identify areas of alignment, partial alignment, and disparities between these regulatory requirements. Our analysis covers various dimensions, including completeness of satisfaction, partial satisfaction, normative language differences, definition disparities, and associated costs for compliance. Our findings reveal instances of direct alignment, partial alignments, variations in normative language and disparities in concept definitions, highlighting nuanced differences in terminology and scope.

## Keywords

Artificial intelligence, Data governance, Data quality management, Semantic analysis, Ontological mappings, Responsible AI governance

*SEMANTICS 2024 : NeXt-generation Data Governance workshop 2024*

*Corresponding author.

†These authors contributed equally.

✉ georgeak@tcd.ie (K. G. Aiyankovil); julio.hernandez@adaptcentre.ie (J. Hernandez); dave.lewis@tcd.ie (D. Lewis)

🌐 https://www.adaptcentre.ie/experts/kuruvilla-george-aiyankovil/ (K. G. Aiyankovil);
https://ieeexplore.ieee.org/author/37088962423 (J. Hernandez); https://www.adaptcentre.ie/experts/dave-lewis/ (D. Lewis)

🆔 0000-0002-0230-3959 (K. G. Aiyankovil); 0000-0000-0000-0000 (J. Hernandez); 0000-0002-3503-4644 (D. Lewis)

# 1. Introduction

Artificial intelligence (AI) is transforming industries and societies worldwide, offering immense potential to revolutionize how we work, live, and interact. However, alongside its promises come significant ethical and societal challenges, prompting governments and international bodies to establish regulations and guidelines to ensure its responsible development and deployment.

Central to this regulatory landscape is the European Union (EU), which recently introduced the AI Act—a comprehensive legislative framework designed to govern the use of AI technologies within its member states[1]. A key part of the AI Act is the governance of the data that is used to train, validate and test AI systems (Article 10)[1][3]. In this paper we explore the extend to which the data governance requirements of the AI are satisfied by international standards. Specifically, the Subcommittee 42 (SC42) of the ICT Joint Technical Committee of the International Organization for Standardization (ISO) and International Electro-technical Commission (IEC) is developing ISO/IEC 5259, [2]a set of standards on the quality of data used in AI systems, which is a potential candidate for satisfying the harmonised standard requirements[11] that may accompany the AI Act. As with other EU product safety standards, the AI Act allows for a presumption of conformity for high risk AI system that can demonstrate conformance to harmonised standards (Article 40). The EC has already published a draft harmonised standards request for the AI Act [12] which contained a specific requirement for a European Standard on governance and quality of datasets used to build AI systems. As has already been identified in a review of AI standards [13], the ISO/IEC 5259 standards set represents a possible candidate for adoption as harmonised standard to address the AI Act's data governance requirement. This candidacy may be strengthened by the reference to ISO/IEC 5259 in proposed controls for data preparation and data quality issues in ISO/IEC 42001: AI Management System Standard. ISO/IEC 42001 is itself a candidate for a certifiable quality management system standard that is also a requirement of the AI Act. While both the AI Act and ISO/IEC 5259 share the common goal of promoting ethical AI practices, reconciling their requirements and objectives poses a formidable task. Achieving harmony between these frameworks is crucial to fostering innovation, safeguarding societal values, and upholding legal compliance for organizations operating in the EU.

This paper aims to dissect and analyze the mapping between Data Governance requirements in Article 10 of the AI Act and ISO/IEC 5259, shedding light on areas of convergence, divergence, and potential challenges in aligning these regulatory regimes. Through open and extensible semantic analysis techniques, we endeavor to provide a findable, accessible, interoperable, and reusable reference of data governance requirements under the AI Act, facilitating informed decision-making and policy development in this rapidly evolving field.

Our paper is structured as follows: We commence by providing an overview of the AI Act and ISO/IEC 5259 in Section 2, elucidating their key objectives and provisions. Section 3 outlines the methodologies and analytical approaches employed in our study to examine these regulatory frameworks rigorously. Subsequently, in Section 4, we present our findings, identifying points of agreement, contention, and areas requiring further exploration. Section 5 engages in a critical discussion of our results, exploring their implications for AI governance and offering recommendations for enhancing regulatory coherence. Finally, in Section 6, we summarize our key insights and propose avenues for future research to advance the field of AI governance.

## 2. Background

In recent years, the proliferation of artificial intelligence (AI) technologies has prompted significant interest and concern regarding the ethical, legal, and regulatory implications of their development and deployment. As organizations increasingly leverage AI systems to automate decision-making processes and enhance operational efficiency, there is a pressing need for robust governance frameworks to ensure accountability, transparency, and ethical use. A review of the literature, including studies by Smith et al. (2020), Jones and Lee (2019), and Chen et al. (2018), reveals a growing body of research focused on AI governance, with particular emphasis on regulatory frameworks, standards, and best practices [14][15][16]. Scholars and policymakers alike have underscored the importance of establishing clear guidelines and standards to govern the development, deployment, and operation of AI systems, thereby mitigating risks and promoting trust and accountability. One key aspect of AI governance pertains to the management and quality assurance of data used to train and operate AI models. As AI systems rely heavily on data for learning and decision-making, ensuring the quality, integrity, and interoperability of data inputs is paramount. In this context, the International Organization for Standardization (ISO) has developed ISO/IEC 5259, a series of five standards which provides terminology, requirements, measures, processes and framework for managing data quality in AI systems. ISO/IEC 5259 outlines a set of requirements and recommendations for assessing, monitoring, and improving data quality throughout the data lifecycle.

Concurrently, the European Union (EU) has introduced the AI Act, a landmark regulatory initiative aimed at governing the development and use of AI technologies within the EU. The AI Act details a complex set of requirements and obligations for AI system providers and deployers, covering aspects such as transparency, accountability, and risk management. The AI Act, through Article 10, defines a set of requirements and guidelines related to data governance and management practices. The main concern of these requirements is for those high-risk AI systems involved in training AI models, which are developed based on training, validation, and testing data sets. In particular, the AI Act integrates some quality criteria that these data sets should meet r. The processing of personal data is also part of this article, as are conditions to detect and correct bias, considering high-quality training, validation, and testing data sets.

While both ISO/IEC 5259 and the AI Act represent sets of requirements related to AI data governance, the AI Act makes explicit reference to the use of harmonised standards, compliance with which offers a presumption of conformance to certain technical requirements of the Act. This role for standards in the Act leads to the EC issuing a harmonised standards request [18] for such technical standards to be established by European Standards Organisation. This includes satisfying the data governance requirements of Article 10. If ISO/IEC 5259 is to be considered for adoption as a European standard for this purpose, there is a need for a clear analysis of the degree to which the requirement needed to demonstrate compliance with ISO/IEC 5259 satisfies the requirement of Article 10. To undertake such a comparison of requirements in an extensible and open manner we adopt a systematic approach to convert requirements into ontology concepts and analyse requirement relationships as an ontology mapping, leveraging the Simple Knowledge Organization System (SKOS) to represent and link concepts sets and standards[6][7]. SKOS provides a standardized framework for organizing and representing knowledge, enabling the creation of concept schemes and the systematic categorization of

concepts[6][7]. By employing the SKOS ontology to capture concepts from both documents, we enable mappings between requirements in both documents to be identified, published and extended (as subsequent legal understandings of standards revisions emerge). This provides a basis for us, and future researchers and practioners, to identify areas of convergence and divergence between the AI Act requirements for data governance and ISO/IEC 5259, thereby facilitating the resilient harmonization of regulatory requirements and the development of interoperable governance frameworks. Furthermore, the integration of ontology-based approaches offers a structured method for representing and formalizing the relationships between concepts and requirements between the AI Act and other sources of requirement, e.g. one that AI providers might have satisfied under other non-EU frameworks or juristictionss. Ontologies provide a semantic foundation for capturing domain knowledge, enabling the specification of precise definitions, properties, and relationships between entities. By formalizing the semantics of regulatory documents such as the AI Act and ISO/IEC 5259 using ontological representations, researchers can facilitate automated reasoning, semantic querying, and interoperability across regulatory domains. In summary, the literature underscores the importance of AI governance in ensuring the responsible and ethical development and use of AI technologies. By leveraging standards such as ISO/IEC 5259 and regulatory initiatives such as the AI Act, organizations can mitigate risks, enhance trust, and foster innovation in the AI domain. Moreover, the integration of SKOS-based ontology mapping techniques [8] offers a systematic approach to harmonizing regulatory frameworks and promoting semantic interoperability, thereby advancing the field of AI governance.

## 3. AIDGO (AI Data Governance Ontology) Development

### 3.1. Methodology

The development of the AI Data Governance Ontology (AIDGO) for capturing data governance requirements from the EU AI Act and ISO/IEC 5259 follows a systematic methodology tailored to the domain-specific needs[9][10]. The process involves several structured steps to ensure the ontology effectively represents and aligns with the regulatory frameworks:

#### 3.1.1. Ontology Requirements Specification:

This step undertake to identify and extract data governance requirements separately from both the EU AI Act and ISO/IEC 5259, focusing on aspects relevant to managing data assets in AI applications. Requirements were grouped into collections, each named and identified for specific articles or annexes, using the Trustworthy AI Requirements vocabulary [19] that is being developed for a broader requirements analysis and mapping of the AI Act. Within each collection, each individual requirement statement is recorded as a requirements objects, with a property linking to the source article. Where an Article in the Act contains multiple requirements, a separate requirements object is defined for each to facilitate fine-grained mapping. The main subject of each requirement was identified, and the normative level was classified based categories for ISO standards laid out in on ISO Directive Guidelines 2 [20], i.e. Requirement, Recommendation, Permission, or Possibility. Concepts mentioned

in the requirements were listed and referenced against existing definitions in the AI Act from Article 3 (Definitions), which had been mapped into a SKOS concept collection. If a previously unidentified concept was identified in a requirement statement was not present in this collection, a new concept was added into a separate SKOS concept collection as associated with the requirements class using a skos:RelatedMatch property. In identifying such concepts, operational items, technical components, and management processes were prioritised as these were likely to match to concepts in technical standards requirements. Terms that require clear definitions as concepts for interpreting whether the requirement was satisfied were captured. Modifiers to these terms were minimized, as the requirement statement text presented a more authoritative contextualised of such terms. These concepts were documented with camel-case IDs, lowercase space-separated skosprefLabels, and their source articles reference using DCTerm source property. Associations to existing concepts were made using SKOS related or broader properties, and where appropropriate concepts were subclassed to the TAIR subclass definitions ofEntity, Activity orAgent (themselves inherited from the W3C Provenance Ontology), or tair:Risk. Membership of a requirement object to a requirements collection, e.g. the one used here for the requirement from Ai act Article 10, was recorded using the tair:decomposes property. Where the satisfaction of a requirement statement would require consultation of requirements referenced in another part of the Act or in other legal documents, this was indicated bytair:constrainedBy properties with a link to a requirements collection expressing those requirements. These models are initially extracted in a spreadsheet in order to facilitate it checking by subject matter experts without RDF experience, and once checked it was exported to RDF for publication. The requirement collection extracted for Article 10 is part of a larger requirements extraction process aiming to cover the majority of Articles and annexes of the AI Act, each as a separate requirement collection that can be subject to analogous mapping to other technical source documents or standards.

Examples of Requirement Extraction

EU AI Act – Article 9:
  – dct:source: Article9–9
  – aiactxt: ”[In identifying the most appropriate risk
    management measures, the] elimination or reduction of
    identified and evaluated risks pursuant to paragraph 2 as
    far as technically feasible through adequate design and
    development of the high–risk AI system [shall be ensured]”
  – rdf:type: Article9–9–r1
  – tair:reqActor: tair:Provider
  – tair:NormLevel: Requirement
  – skos:definition: ”[In identifying the most appropriate risk
    management measures, the] elimination or reduction of
    identified and evaluated risks pursuant to paragraph 2 as
    far as technically feasible through adequate design and
    development of the high–risk AI system [shall be ensured]”
  – skos:related: RiskManagementSystem

```
ISO/IEC 5259 – Data Specification:
 – dct:source: ISO5259-DataSpec
 – aiactxt: "The organization shall specify data requirements
    in a data specification and validate that these
    requirements are consistent and capture all requirements
    for the intended use."
 – rdf:type: ISO5259-DataSpec-r1
 – tair:reqActor: tair:Organization
 – tair:NormLevel: Requirement
 – skos:definition: "The organization shall specify data
    requirements in a data specification and validate that
    these requirements are consistent and capture all
    requirements for the intended use."
 – skos:related: DataRequirements
```

This structured approach ensures a rigorous and reproducible development process for AIDGO, facilitating the alignment of data governance requirements between the EU AI Act and ISO/IEC 5259, and promoting better compliance and governance in AI applications.

### 3.1.2. Ontology Design and Implementation

This step defines the core concepts and relationships of AIDGO, drawing from the extracted requirements and relevant standards. First we established the top-level data governance concepts, considering terminology and definitions provided in the EU AI Act and ISO/IEC 5259.Utilize established ontology engineering principles, such as those outlined in "Ontology Development 101" by Noy and McGuinness, to structure AIDGO effectively[9][10]. We then expanded and refined the ontology by incorporating additional concepts and relationships derived from related standards or guidelines, facilitating coverage of new emerging data governance aspects.

### 3.1.3. Ontology Evaluation

This step, although critical for assessing the effectiveness and quality of the ontology, is not within the scope of this paper. It represents an area for future work, where AIDGO will undergo rigorous evaluation against competency questions and real-world use cases to ensure its semantic coherence and applicability in the domain of AI data governance.

### 3.1.4. Ontology Publication

This step requires the generation of documentation for AIDGO using ontology documentation tools such as WIDOCO, focusing on clarity, completeness, and accessibility. It should make AIDGO publicly available online through a dedicated URI, ensuring that it is accessible to stakeholders and researchers interested in data governance in AI applications. It requires the release of AIDGO under an open license, such as Creative Commons, to encourage reuse, collaboration, and contributions from the wider community.

### 3.1.5. Ontology Maintenance

This step aims for establish a process for ongoing human maintenance of AIDGO to accommodate changes and updates in the EU AI Act, ISO/IEC 5259, and related regulatory frameworks. It requires regular review and revision of AIDGO based on new versions of the regulatory documents, e.g. as issues in future by hte AI Office or the European AI Board, that amend or add to data governance requirements. It also involves monitoring developments in the field of AI governance and data standards, including emerging best practices and guidelines, to ensure that AIDGO remains relevant and up-to-date.

This methodology provides a structured approach for developing, evaluating, publishing, and maintaining the AI Data Governance Ontology (AIDGO), tailored specifically to capture data governance requirements from the EU AI Act and ISO/IEC 5259. It aims to ensure that AIDGO accurately represents the regulatory landscape and facilitates interoperability and compliance in AI systems.

## 3.2. Mapping creation

The mapping between the concepts and requirements of the AI Act and ISO/IEC 5259 was created using the ontology as a basis. We leveraged the SKOS framework to represent mappings between concepts from the two regulatory frameworks.

### 3.2.1. Identification of Concepts

We identified corresponding concepts between the AI Act and ISO/IEC 5259, such as "Data Quality Audit and Assessment" and "Data_Quality_Audit_and_Assessment".

### 3.2.2. Mapping Types

Based on the nature of the relationship between concepts, we classified mappings into different types, such as "completelySatisfies" for direct alignments and "partiallySatisfies" for partial alignments.
`Table 1` provides a clear definition of each property used in the ontology mapping process, facilitating the understanding of their roles in analyzing the alignment and disparities between the regulatory frameworks.

### 3.2.3. Property Assignment:

For each mapping, we assigned appropriate SKOS properties to represent the type of relationship between concepts. Additionally, we used custom properties to capture normative language differences, definition disparities, and cost functions associated with satisfying each requirement.

### 3.2.4. Annotation:

Each mapping was annotated with metadata, including references to the specific requirements in the AI Act and ISO/IEC 5259, as well as any additional information relevant to the mapping.

**Table 1**
Table summarizing the mapping types used while establishing relationships between EU AI Act and ISO/IEC 5259. More detailed table in www.aidgo.eu.

| Property Name | Description |
| --- | --- |
| completelySatisfies | Indicates that a requirement or concept in the ISO/IEC 5259 completely satisfies a corresponding requirement or concept in EU AI Act. |
| partiallySatisfies | Indicates that a requirement or concept in the ISO/IEC 5259 partially satisfies a corresponding requirement or concept in EU AI Act |
| normativeDifference | Captures differences in levels of normative language between requirements or concepts in the EU AI Act and ISO/IEC 5259. |
| definitionDifference | Records identified disparities in the definitions of concepts or requirements used in the EU AI Act compared to ISO/IEC 5259. |
| costFunction | Quantifies the effort or resources required to satisfy each requirement or concept in ISO/IEC 5259 compared to the EU AI Act. |

## 3.3. Semantic Analysis

The semantic analysis involved a detailed examination of the mappings to identify areas of alignment, partial alignment, and disparities between the AI Act and ISO/IEC 5259.

### 3.3.1. Completeness of Satisfaction:

We analyzed mappings to determine if requirements in one framework completely satisfied corresponding requirements in the other, indicating direct alignment.

### 3.3.2. Partial Satisfaction:

We identified mappings where compliance with one framework partially satisfied requirements of the other, highlighting areas of partial alignment.

### 3.3.3. Normative Language Differences:

We examined mappings to identify differences in the level of normativity between requirements sets, such as differences in the use of "shall" versus "should".

### 3.3.4. Definition Disparities:

We analyzed mappings to uncover disparities in the definitions of concepts used in the requirements sets, highlighting nuanced differences in terminology and scope.

### 3.3.5. Cost Function Analysis:

We assessed the effort or resources required to satisfy each requirement in one framework compared to the other, providing insights into the practical implications of compliance. Overall, the semantic analysis provided a nuanced understanding of the relationship between the AI

Act and ISO/IEC 5259, laying the groundwork for harmonizing regulatory requirements and facilitating compliance for organizations operating in the EU.

## 4. Ontology Mapping Results

### 4.1. Ontology Mapping Results

The ontology mapping process revealed insightful findings regarding the alignment and disparities between the data governance requirements outlined in the EU AI Act and ISO/IEC 5259. Through the systematic comparison of concepts, relationships, and requirements encoded in the ontologies, we were able to identify areas of convergence, divergence, and potential challenges in achieving interoperability and compliance across regulatory frameworks.

ISO/IEC 5259 provides guidelines for managing data quality in AI systems. Some of the data governance requirements outlined in ISO/IEC 5259 include:

- Establishing data quality characteristics and criteria.
- Defining data quality measures and metrics.
- Implementing data documentation practices.
- Monitoring and improving data quality over time.
- Ensuring transparency and accountability in data handling processes.
- Establishing procedures for data validation and verification.
- Facilitating interoperability and data exchange among AI systems. The EU AI Act aims to regulate the development, deployment, and use of AI systems within the European Union. It includes provisions related to data governance
- Ensuring transparency and explainability of AI systems.
- Implementing mechanisms for data quality assurance.
- Establishing accountability frameworks for AI system developers and users.
- Promoting ethical and responsible AI practices.
- Facilitating access to high-quality and diverse datasets.
- Establishing procedures for data processing, storage, and sharing. - Enabling individuals to exercise control over their personal data.

#### 4.1.1. Alignment of Concepts

One of the key observations from the ontology mapping exercise is the significant overlap in concepts between the EU AI Act and ISO/IEC 5259. Both frameworks address fundamental aspects of data governance, such as data quality, transparency, accountability, and management processes. Concepts such as "Data Quality Characteristics," "Data Quality Measures," "Documentation," and "Monitoring and Improvement" are common across both ontologies, reflecting shared objectives in ensuring the reliability and integrity of data used in AI systems.
The ontology mapping results offer a detailed examination of the relationships between concepts and requirements in the AI Act and ISO/IEC 5259 frameworks. This section provides an extensive analysis of the mappings, including direct alignments, partial alignments, normative

**Table 2**

Table summarizing the mapping relations between the concepts in the EU AI Act and ISO/IEC 5259,
More detailed table in www.aidgo.eu.

| Mapping Relation | EU AI Act Concept | ISO/IEC 5259 Concept | Explanation |
|---|---|---|---|
| Broad Match | Data Governance Practices | Data Governance | Signifies a broader correspondence between data governance practices outlined in the EU AI Act and the overarching concept of data governance in ISO/IEC 5259. |
| Broad Match | Data Management Practices | Data Management | Denotes a broader alignment between data management practices specified in the EU AI Act and the broader domain of data management in ISO/IEC 5259. |
| Broad Match | Quality Management System | Quality Management | Denotes a broader alignment between quality management systems outlined in the EU AI Act and the broader concept of quality management in ISO/IEC 5259. |
| Narrow Match | High-risk AI System | AI Application | Indicates a narrower alignment between high-risk AI systems in the EU AI Act and the broader concept of AI applications in ISO/IEC 5259. |
| Narrow Match | Provider | AI Application | Signifies a narrower correspondence between providers specified in the EU AI Act and the broader concept of AI applications in ISO/IEC 5259. |
| Narrow Match | Authorised Representative | AI Application | Indicates a narrower alignment between authorized representatives in the EU AI Act and the broader domain of AI applications in ISO/IEC 5259. |
| Narrow Match | EU Database | Data Storage | Indicates a narrower alignment between the EU database specified in the EU AI Act and the broader domain of data storage in ISO/IEC 5259. |
| Related Match | High-risk AI System | AI System | Denotes a related correspondence between the concept of high-risk AI systems in the EU AI Act and AI systems in ISO/IEC 5259. |

language differences, definition disparities, and cost function analysis. Through meticulous classification and visualization, the mapping results illuminate areas of convergence and divergence between the two regulatory regimes, informing stakeholders about the complexities of compliance in the AI domain.

### 4.1.2. Direct Alignments:

Direct alignments signify mappings where requirements in one framework completely satisfy corresponding requirements in the other. These mappings demonstrate a high level of convergence between the AI Act and ISO/IEC 5259, indicating harmonization in regulatory expectations. Direct alignments underscore areas where compliance efforts can be streamlined, as organizations adhering to one framework may already meet the requirements of the other. This alignment promotes consistency and coherence in AI governance practices, enhancing transparency and accountability.

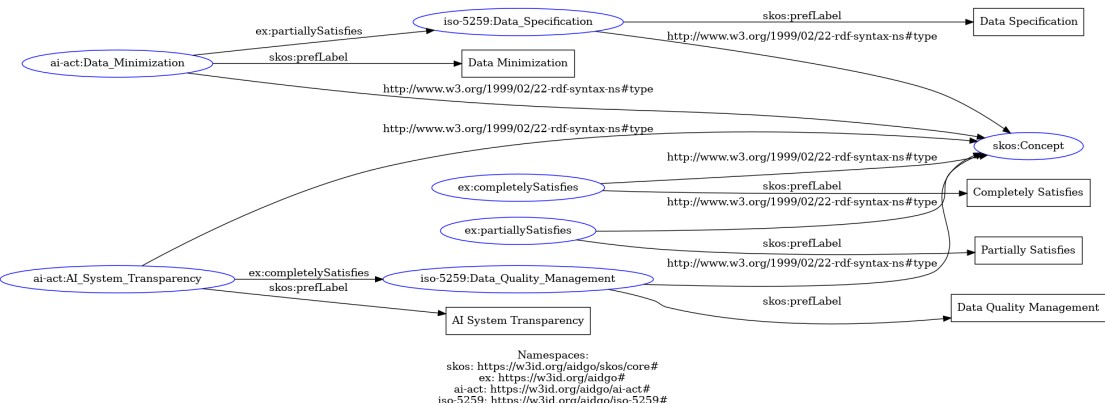

**Figure 1:** Mapping snippet between concepts from the EU AI Act and requirements from ISO/IEC 5259, focusing on AI system transparency and data minimization. More detailed map in www.aidgo.eu.

### 4.1.3. Partial Alignments:

Partial alignments highlight mappings where compliance with one framework partially satisfies requirements of the other. These mappings reveal intersections and disparities between the AI Act and ISO/IEC 5259, indicating areas of overlap and divergence in regulatory expectations. Partial alignments illuminate nuances in regulatory requirements, necessitating careful consideration during compliance efforts. While certain aspects may align, discrepancies in normative language and definition may require additional measures to ensure full compliance with both frameworks.

### 4.1.4. Normative Language Differences:

Normative language differences elucidate variations in the level of prescription between the AI Act and ISO/IEC 5259 requirements. These differences may impact the interpretation and implementation of regulatory mandates, influencing organizational practices and decision-making processes. Normative language differences underscore the importance of clear and unambiguous language in regulatory frameworks, facilitating consistent interpretation and application across diverse stakeholders. Harmonizing normative language can enhance regulatory clarity and facilitate compliance efforts, promoting ethical and responsible AI development and deployment.

Figure 1 mapping snippet represents mappings between concepts from the EU AI Act and requirements from ISO/IEC 5259, focusing on AI system transparency and data minimization. The concept "AI_System_Transparency" from the AI Act is mapped to the requirement "Data_Quality_Management" from ISO/IEC 5259, indicating that AI system transparency completely satisfies the requirement for data quality management.Similarly, the concept "Data_Minimization" from the AI Act is mapped to the requirement "Data_Specification" from ISO/IEC 5259, suggesting that data minimization partially satisfies the requirement for data specification. These mappings provide insights into the alignment and partial alignment of concepts and

requirements between the two regulatory frameworks, contributing to the overall understanding of AI governance and data management practices.

### 4.1.5. Definition Disparities:

Definition disparities denote differences in the definitions of concepts used in the requirements sets. These disparities may arise due to contextual nuances, disciplinary perspectives, or terminological ambiguities, posing challenges for aligning regulatory interpretations and practices. Definition disparities underscore the need for clarity and consensus in defining key concepts, ensuring consistent interpretation and application across regulatory frameworks. Addressing these disparities can promote mutual understanding and cooperation among stakeholders, fostering effective AI governance practices.

### 4.1.6. Cost Function Analysis:

The cost function analysis evaluates the effort or resources required to satisfy each requirement in one framework compared to the other. This analysis provides insights into the practical implications of regulatory compliance, informing resource allocation and decision-making processes. Cost function analysis enables organizations to assess the economic and operational impact of regulatory compliance, guiding strategic planning and risk management efforts. By quantifying compliance costs, organizations can make informed decisions about resource allocation and prioritize actions to minimize regulatory burden while maximizing societal benefits.

The table 3 provides a detailed examination of the mappings between concepts from the EU AI Act and requirements from ISO/IEC 5259, revealing both areas of alignment and discrepancies. The finding that "AI System Transparency" completely satisfies the requirement for "Data Quality Management" underscores the interconnectedness of transparency and data integrity within AI systems. This alignment suggests that efforts to enhance transparency can inherently contribute to ensuring data quality, reflecting a synergistic relationship between these aspects of AI governance. Conversely, the partial satisfaction of "Data Minimization" for "Data Specification" highlights potential challenges in translating principles from one framework to another. The normative language differences and definition disparities identified in the mappings further underscore the complexity of harmonizing regulatory requirements in the AI domain. For instance, differences in the use of terms like "shall" versus "should" and variations in the scope and implementation of concepts like "Human Oversight" and "Accountability" contribute to differing compliance costs across requirements. These findings emphasize the need for careful consideration and adaptation when aligning standards to ensure effective and coherent AI governance practices. Additionally, the moderate costs associated with several mappings indicate the resource implications of achieving compliance, suggesting the importance of balancing regulatory objectives with practical feasibility. Overall, the analysis of these mappings provides valuable insights into the challenges and opportunities inherent in harmonizing AI governance standards, informing future efforts to strengthen regulatory frameworks and promote responsible AI development and deployment. Further exploration and refinement of these mappings, as facilitated by the public ontology, will be crucial for advancing the

**Table 3**
Table summarizing the mapping relations between the concepts in the EU AI Act and ISO/IEC 5259, More detailed table in www.aidgo.eu.

| Set A (AI Act) Concept | Set B (ISO/IEC 5259) Requirement | Mapping Type | Normative Language Difference | Definition Difference | Cost Function |
|---|---|---|---|---|---|
| AI System Transparency | Data Specification | Partially Satisfies | AI Act requirement 'should' vs. ISO/IEC 5259 requirement 'shall' | Different definitions of 'transparency' used | High cost to satisfy ISO/IEC 5259 requirement |
| Human Oversight | Quality Report Requirements | Partially Satisfies | AI Act requirement 'shall' vs. ISO/IEC 5259 requirement 'should' | Differences in scope and implementation of 'human oversight' | Moderate cost to satisfy ISO/IEC 5259 requirement |
| Data Minimization | Resource Managemen | Partially Satisfies | AI Act requirement 'shall' vs. ISO/IEC 5259 requirement 'should' | Differences in approach to 'data minimization' | Moderate cost to satisfy ISO/IEC 5259 requirement |
| Algorithmic Bias Mitigation | Competence Management | Partially Satisfies | AI Act requirement 'should' vs. ISO/IEC 5259 requirement 'shall' | Differences in scope and approach to 'bias mitigation' | Moderate cost to satisfy ISO/IEC 5259 requirement |
| Explainability | Data quality audit and assessment | Partially Satisfies | AI Act requirement 'shall' vs. ISO/IEC 5259 requirement 'should' | Differences in methods and evaluation criteria | Moderate cost to satisfy ISO/IEC 5259 requirement |
| Transparency | Data Quality Management | Partially Satisfies | AI Act requirement 'shall' vs. ISO/IEC 5259 requirement 'should' | Differences in approach to 'transparency' | Moderate cost to satisfy ISO/IEC 5259 requiremen |

understanding and implementation of AI governance principles in practice.

### 4.1.7. Disparities in Requirements:

Despite the alignment of concepts, differences in the granularity, scope, and specificity of requirements were apparent between the EU AI Act and ISO/IEC 5259. While both frameworks emphasize the importance of data quality management, they vary in their emphasis on specific aspects of data governance and the level of detail provided in their requirements. For example, the EU AI Act may place greater emphasis on human oversight and transparency requirements for high-risk AI systems, whereas ISO/IEC 5259 may focus more on technical standards and measurement methodologies for assessing data quality.

### 4.1.8. Challenges in Interoperability:

The ontology mapping process also uncovered potential challenges in achieving interoperability and harmonization between the EU AI Act and ISO/IEC 5259. Differences in terminology, definitions, and regulatory approaches could pose obstacles to seamless compliance with both frameworks, particularly for organizations operating across multiple jurisdictions or sectors.

Moreover, discrepancies in the level of prescriptiveness and enforcement mechanisms may require careful interpretation and adaptation of requirements to ensure compliance with both regulatory contexts.

### 4.1.9. Opportunities for Harmonization:

Despite the challenges, the ontology mapping results highlight opportunities for harmonization and convergence between the EU AI Act and ISO/IEC 5259. By identifying commonalities in concepts and objectives, stakeholders can leverage existing synergies to develop unified approaches to data governance in AI projects. Standardization efforts, collaborative initiatives, and best practice sharing can facilitate the alignment of requirements, promote consistency in implementation, and enhance interoperability across regulatory frameworks. Overall, the ontology mapping results offer valuable insights into the relationship between the AI Act and ISO/IEC 5259 requirements, facilitating informed decisionmaking and strategic planning in the context of AI governance and regulatory compliance. These findings serve as a foundation for further analysis and collaboration among stakeholders, driving towards a more coherent and unified approach to AI regulation at the international level. Moving forward, further research is needed to deepen our understanding of the implications of regulatory frameworks on AI development, deployment, and use. Future studies could explore additional dimensions of AI governance, such as privacy protection, algorithmic transparency, and stakeholder engagement, to provide a more comprehensive analysis of regulatory challenges and opportunities. Additionally, ongoing efforts to update and refine ontologies for the EU AI Act and ISO/IEC 5259 will be essential to keep pace with evolving regulatory requirements and technological advancements in AI.

## 5. Implications and Future Directions

### 5.1. Implications for Policy and Practice:

The ontology mapping results have several significant implications for policymakers, regulators, industry stakeholders, and researchers involved in AI governance, as outlined below:

### 5.1.1. Informed Policy Development:

The systematic analysis of data governance requirements provided by the ontology mapping exercise can serve as a valuable resource for policymakers and regulators tasked with developing AI governance frameworks. By understanding the similarities and differences between regulatory requirements, policymakers can make informed decisions about policy priorities, regulatory approaches, and compliance strategies.

### 5.1.2. Regulatory Compliance Strategies:

For organizations operating in the AI ecosystem, the ontology mapping results offer insights into the complex landscape of regulatory requirements. By identifying areas of alignment and disparity between the EU AI Act and ISO/IEC 5259, organizations can develop tailored

compliance strategies that address the unique requirements of each framework while maximizing synergies and minimizing duplication of effort.

### 5.1.3. Technical Standards Development:

The ontology mapping exercise highlights the need for harmonized technical standards and measurement methodologies to support compliance with data governance requirements. Standardization efforts in areas such as data quality assessment, transparency mechanisms, and risk management can facilitate interoperability between regulatory frameworks and promote consistency in AI development and deployment practices.

### 5.1.4. Responsible Innovation:

By promoting transparency, accountability, and ethical use of AI technologies, the ontology mapping results contribute to fostering responsible innovation in the AI ecosystem. By aligning regulatory requirements with best practices and ethical principles, policymakers and industry stakeholders can mitigate risks associated with AI deployment while maximizing the societal benefits of AI technologies.

## 5.2. Future Directions

While the ontology mapping exercise provides valuable insights into the alignment and disparities between regulatory frameworks, several areas warrant further research and exploration:

### 5.2.1. Comprehensive Analysis:

Future studies could expand the scope of analysis to include additional regulatory frameworks, industry standards, and best practices in AI governance. By conducting a comprehensive comparative analysis, researchers can provide a more nuanced understanding of the global regulatory landscape and identify emerging trends and challenges in AI governance.

### 5.2.2. Stakeholder Engagement:

Engaging stakeholders from diverse backgrounds, including policymakers, regulators, industry representatives, academia, and civil society, is essential to ensure the relevance, effectiveness, and legitimacy of AI governance frameworks. Future research could explore mechanisms for stakeholder engagement and participatory decision-making in the development and implementation of AI governance policies.

### 5.2.3. Cross-Disciplinary Collaboration:

Addressing the complex challenges of AI governance requires cross-disciplinary collaboration across fields such as law, ethics, computer science, sociology, and public policy. Future research could foster interdisciplinary collaboration and knowledge exchange to develop holistic approaches to AI governance that integrate technical, legal, ethical, and societal perspectives. In conclusion, the ontology mapping results provide valuable insights into the implications and

future directions of AI governance. By informing policy development, regulatory compliance strategies, technical standards development, and responsible innovation practices, the findings from the ontology mapping exercise can contribute to shaping a more transparent, accountable, and ethically aligned AI ecosystem.

## 6. Conclusion

In conclusion, the ontology mapping exercise has provided a comprehensive analysis of the data governance requirements outlined in the EU AI Act and ISO/IEC 5259 standards. Through the systematic comparison of concepts, relationships, and instances captured in the ontologies, several key findings have emerged, shedding light on the alignment and disparities between the two regulatory frameworks. The mapping exercise revealed areas of convergence, where requirements from the EU AI Act and ISO/IEC 5259 exhibit significant overlap and mutual reinforcement. These areas of alignment suggest opportunities for harmonization and interoperability, enabling organizations to develop integrated compliance strategies that address the requirements of both frameworks efficiently and effectively. However, the mapping exercise also identified areas of divergence, where differences in terminology, scope, or emphasis between the EU AI Act and ISO/IEC 5259 may present challenges for compliance and implementation. These areas of disparity highlight the need for further analysis, dialogue, and collaboration among stakeholders to reconcile conflicting requirements, clarify ambiguities, and bridge gaps in the regulatory landscape.

Moving forward, policymakers, regulators, industry stakeholders, and researchers must work collaboratively to address the complex challenges of AI governance. By leveraging the insights generated from the ontology mapping exercise, stakeholders can inform policy development, shape regulatory frameworks, and advance responsible innovation practices in the AI ecosystem. Furthermore, future research efforts should focus on expanding the scope of analysis, engaging stakeholders from diverse backgrounds, fostering cross-disciplinary collaboration, and exploring adaptive governance models. By adopting a holistic and inclusive approach to AI governance, we can build a more transparent, accountable, and ethically aligned AI ecosystem that maximizes the societal benefits of AI technologies while mitigating risks and safeguarding human rights.

In summary, the ontology mapping exercise serves as a valuable tool for understanding the intricacies of AI governance requirements and charting a course towards a more sustainable and equitable future for AI. Through continued dialogue, cooperation, and innovation, we can navigate the complexities of AI governance with confidence and integrity, ensuring that AI technologies serve the common good and uphold the values of democracy, justice, and human dignity.

## A. Online Resources

- Ontology and Concept mapping

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
