# OpenReview forum: "Mapping Data Governance Requirements Between the European Union’s AI Act and ISO/IEC 5259: A Semantic Analysis"
_SEMANTiCS.cc/2024/Workshop/NXDG — NXDG 2024_

### Official Review · ~Anelia_Kurteva1 · 2024-07-24
**Timely work that tries to discover the alignments and disparities between the AI Act and ISO 5259. The paper needs significant imporvement in terms of writing and formating to ease the reader.**

**Rating:** 6
**Confidence:** 4

**Review:**

This is an interesting and timely work that tries to discover the alignments and disparities between the AI Act and ISO 5259. The authors discuss their work in detail, provide examples, derive conclusions and discuss future work.

At times the paper is difficult to follow and understand. My comments are as follows:

The last paragraph in section 2 (is too long and mixes up different topics) - instead structure it into several paragraphs dedicated to their corresponding topics

The benefits of using semantics are clear. The authors can also provide specific references (as examples of work) when mentioning these benefits at the end of section 2.

What specific "SKOS-based ontology mapping techniques" are the authors referring to?

Sometimes references are added at the end of the sentence and it's unclear what they should reference.

The methodology (section 3.1) should clearly state the steps and then each step can be elaborated on in a dedicated subsection.

Some sentences in section 3.2 would benefit from paraphrasing and simplification.

Section 3.1.3 is missing information on methods the authors plan to use to evaluate the ontology in terms of semantics and structure (e.g. with reasoners). Currently only use case (application) evaluation has been mentioned. Further, it would be nice to see a discussion about the authors' background and their legal expertise since the mapping and the ontology will represent their views. Has a legal expert been involved and how do the authors plan to validate their analysis and work regarding this? Who derived and how the competency questions?

Tables have not been referenced in the text. The table 2's caption can also explain the difference between the match categories.

The methodology can be visualised with a simple graph to better guide the reader.

Minor comments:

Missing citation of the AI Act

Missing space between words and references and after the end of sentences

Section 2, "In particular, the AI Act integrates some quality criteria that these data sets

Should meet r." - incomplete final sentence word

Missing reference to SKOS after being mentioned in the text

Terms such as TAIR should be defined when first mentioned

Some spelling and grammar issues - "Ai act Article 10", "Item that requires clear definitions for interpreting", "These models are initially extracted"

Overall, the paper's writing style needs improvement in terms of sentence structure and language (grammar and spelling) so that the authors can better convey their ideas to the reader

There is no need for ":" in the title of the sections

Seems as if section 3.3.2 should be part of section 3.3.1

Missing reference to ISO/IEC 5259 in section 4.1 when mentioning examples of requirements/guidelines.

Reference formatting is inconsistent

---

### Official Review · ~Lola_Montero_Santos1 · 2024-07-24
**Great work in progress, a lot of work still needed.**

**Rating:** 5
**Confidence:** 2

**Review:**

# Comments on the substance

The paper indicates its purpose clearly. It has a constrained scope, and it appears to meet the stated goal in connection with this scope. However, this paper needs a lot of work in some sections in terms of formatting, readability, spelling, and grammar. The paper's goal is there, and it is interesting, but the structure of the paper is very much a work in progress.

More detailed comments:

* “the AI Act—a comprehensive legislative framework designed to govern the use of AI technologies within its member states[1]” (p. 2) Many (legal) authors would argue against this statement.

* [https://aidgo.eu/](https://aidgo.eu/) is currently not operational. Readers of this article will want to be able to check it out.

* “Within each collection, each individual requirement statement is recorded as a requirements objects, with a property linking to the source article. Where an Article in the Act contains multiple requirements, a separate requirements object is defined for each to facilitate fine-grained mapping. The main subject of each requirement was identified, and the normative level (Requirement, Recommendation, Permission, or Possibility) was classified based on ISO Directive Guidelines 2 [20].” (p. 4) This sentence is not fully clear for a reader who is not an expert in ontology requirements specification. The same happens with many sentences and paragraphs within Section 3.1.1.

* I really do not understand the “Examples of Requirement Extraction” [Aiyankovil et al., 2024, p. 5]. Does this need to be better explained to a broader audience?

Section “4.1.6. Cost Function Analysis: The cost function analysis evaluates” ([Aiyankovil et al., 2024, p. 12] requires more backing. How are the cost functions calculated? What sources are you using to substantiate whether compliance with one requirement or another entails a High/Moderate cost??

# Spelling issues

There are many spelling errors in this piece. The authors should go over the text in depth and get a spelling/grammar check. I have identified the following:

* “producing an requirement analysis” ([Aiyankovil et al., 2024, p. 1])

* “Through an open and extensible semantic analysis techniques,” ([Aiyankovil et al., 2024, p. 2])

* “we adopt aa systematic” ([Aiyankovil et al., 2024, p. 3)

* “as subsequent legal understanding of standards revisions emerge” ([Aiyankovil et al., 2024, p. 4])

* “one that AI provider” ([Aiyankovil et al., 2024, p. 4)

* “AI Act..” ([Aiyankovil et al., 2024, p. 4)

* “These model” ([Aiyankovil et al., 2024, p. 5])

* “Ai act” ([Aiyankovil et al., 2024, p. 5]) .

* “nological ambiguities,” ([Aiyankovil et al., 2024, p. 12]) This cannot be the beginning of the sentence.

# Readability issues

There are a couple of very convoluted sentences:

* “This candidacy may be strengthened by the reference to ISO/IEC 5259 in proposed controls for data preparation and data quality issues in ISO/IEC 42001, AI Management System, which is a candidate for a certifiable quality management system standard that is also a requirement of the AI Act.” ([Aiyankovil et al., 2024, p. 2)

* “While both ISO/IEC 5259 and the AI Act represent set of requirements related to AI data governance, the requirement of the AI Act for harmonised standards compliance with which offer a presumption of conformance to certain technical requirements of the Act leads to the EC issuing a harmonised standards request [18] for such technical standards to be established by European Standards Organisation which includes satisfying the data governance requirements of Article 10.” ([Aiyankovil et al., 2024, p. 3])

* “Item that requiring clear definitions for interpreting the satisfaction of the requirement satisfaction were captured, and modifiers to these were minimized, as the requirement statement text presented a better contextualised version of this..” ([Aiyankovil et al., 2024, p. 5]) This sentence is really unclear.

Several sentences read as headings or subheadings, not as proper sentences. Is this a formatting error?:

* “Identify and extract data governance requirements separately from both the EU AI Act and ISO/IEC 5259, focusing on aspects relevant to managing data assets in AI applications.” ([Aiyankovil et al., 2024, p. 4])

* “Requirement Identification” ([Aiyankovil et al., 2024, p. 4]) Is this also a subheading?

* “Responsible Innovation:” ([Aiyankovil et al., 2024, p. 15]) Is this a heading?


The following paragraph consists of instructions; it does not read as a proper paragraph: “Define the core concepts and relationships of AIDGO, drawing from the extracted requirements and relevant standards. Establish the top-level data governance concepts, considering terminology and definitions provided in the EU AI Act and ISO/IEC 5259.Utilize established ontology engineering principles, such as those outlined in ”Ontology Development 101” by Noy and McGuinness, to structure AIDGO effectively[9][10]. Expand and refine the ontology by incorporating additional concepts and relationships derived from related standards or guidelines, ensuring comprehensive coverage of data governance aspects.” ([Aiyankovil et al., 2024, p. 6])

* The same occurs in “3.1.4. Ontology Publication” and “3.1.5. Ontology Maintenance”.

---

### Decision · Program_Chairs · 2024-08-02

Accept